# Heterozygous Pathogenic Nonsense Variant in the *ATM* Gene in a Family with Unusually High Gastric Cancer Susceptibility

**DOI:** 10.3390/biomedicines11072062

**Published:** 2023-07-22

**Authors:** Daniele Guadagnolo, Gioia Mastromoro, Enrica Marchionni, Aldo Germani, Fabio Libi, Soha Sadeghi, Camilla Savio, Simona Petrucci, Laura De Marchis, Maria Piane, Antonio Pizzuti

**Affiliations:** 1Department of Experimental Medicine, School of Medicine and Dentistry, Sapienza University of Rome, 00185 Rome, Italyenrica.marchionni@uniroma1.it (E.M.); antonio.pizzuti@uniroma1.it (A.P.); 2Department of Clinical and Molecular Medicine, School of Medicine and Psychology, Sapienza University of Rome, 00185 Rome, Italy; aldo.germani@uniroma1.it (A.G.); soha.sadeghi@uniroma1.it (S.S.); simona.petrucci@uniroma1.it (S.P.); maria.piane@uniroma1.it (M.P.); 3Medical Genetics Unit, Department of Diagnostic Sciences, Sant’Andrea University Hospital, 00189 Rome, Italy; flibi@ospedalesantandrea.it (F.L.); csavio@ospedalesantandrea.it (C.S.); 4Department of Radiological, Oncological and Anatomopathological Science, Sapienza University of Rome, 00185 Rome, Italy; laura.demarchis@uniroma1.it; 5Oncology B Unit, Department of Hematology, Dermatology and Oncology, Policlinico Umberto I Univeristy Hospital, 00161 Rome, Italy

**Keywords:** gastric cancer, ATM, cancer susceptibility

## Abstract

Germline pathogenic variants (PVs) in the Ataxia Telangiectasia mutated (*ATM*) gene (MIM* 607585) increase the risk for breast, pancreatic, gastric, and prostatic cancer and, to a reduced extent, ovarian and colon cancer and melanoma, with moderate penetrance and variable expressivity. We describe a family presenting early-onset gastric cancer and harboring a heterozygous pathogenic *ATM* variant. The proband had gastric cancer (age 45) and reported a sister deceased due to diffuse gastric cancer (age 30) and another sister who developed diffuse gastric cancer (age 52) and ovarian serous cancer. Next generation sequencing for cancer susceptibility genes (*APC*, *ATM*, *BRD1*, *BRIP1*, *CDH1*, *CDK4*, *CDKN2A*, *CHEK2*, *EPCAM*, *MLH1*, *MRE11*, *MSH2*, *MSH6*, *MUTYH*, *NBN*, *PALB2*, *PMS2*, *PTEN*, *RAD50*, *RAD51C*, *RAD51D*, *RECQL1*, *SMAD4*, *STK11*, and *TP53*) was performed. Molecular analysis identified the truncating c.5944C>T, p.(Gln1982*) variant in the *ATM* (NM_000051.3; NP_000042.3) in the proband. The variant had segregated in the living affected sister and in the unaffected daughter of the deceased affected sister. Familial early-onset gastric cancer is an unusual presentation for *ATM*-related malignancies. Individual variants may result in different specific risks. Genotype–phenotype correlations are challenging given the low penetrance and variable expressivity. Careful family history assessments are pivotal for prevention planning and are strengthened by the availability of molecular diagnoses.

## 1. Introduction

The genetic landscape of monogenic cancer susceptibility is varied and complex, with many variants in several genes possibly involved in determining an individual’s susceptibility to different neoplasms. The reduced penetrance for some of these phenotypes and the frequency of sporadic cases can hamper clinical research in this field. In addition, specific risks for individual variants or genes are often not available. Gastric cancer (specifically, gastric adenocarcinoma) is a common and severe neoplasm with poor prognosis, being the fourth worldwide cause of cancer mortality in frequency. Ten percent of gastric cancer cases are familial [1]. The *CDH1* (E-cadherin, MIM *192090) gene is an ascertained and common cause of hereditary diffuse gastric cancer (OMIM #137215), an autosomal dominant condition with high penetrance (up to 76% at 80 years of age) for early-onset diffuse gastric cancer and lobular breast cancer [1]. Pathogenic variants in several other genes can cause gastric cancer susceptibility, usually alongside with risks for other neoplasms, including the *ATM* gene (MIM *607585) [2]. Biallelic *ATM* mutations underlie Ataxia Telangiectasia (AT) (MIM #08900), an autosomal recessive cancer prone disorder with neurological and immunological involvement. The association between cancer susceptibility and pathogenic variants (PVs) in the *ATM* gene was initially detected in AT families, where 10% of AT patients with homozygous or compound heterozygous *ATM* PVs were affected by lymphoma or leukemia, and the female heterozygous carriers presented a higher incidence of breast cancer [3]. This observation was later confirmed in several studies where it was found that women with heterozygous loss-of-function PVs in the *ATM* gene had a 2.3-fold increased risk of breast cancer compared with the general population. This led to the definition of *ATM* as a breast cancer susceptibility gene with moderate penetrance [4,5]. Several studies subsequently reported the association between different types of tumors and PVs in the *ATM* gene [6,7,8,9], including gastric cancer [10]. Estimates of specific cancer risks are emerging, and for gastric cancer, the reported risk conferred by heterozygous *ATM* PVs is up to three times higher than in the general population (odds ratio (OR) of 2.97; 95% CI, 1.66–5.31) [11]. Despite the growing evidence, there is no consensus on the management of gastric cancer risk in individuals and families harboring an *ATM* PV, and gastric cancer screening in *ATM* PV carriers is not included in the surveillance proposed by the current National Comprehensive Cancer Network (NCCN) guidelines^®^ (last update: Version 3.2023—13 February 2023) [12]. We herein report on a family ascertained for hereditary gastric cancer, with a heterozygous pathogenic truncating variant in *ATM*. The frequency of gastric cancer in the family appears to be unusual for kindred individuals with *ATM* mutation carriers.

## 2. Materials and Methods

The proband (II:3) was a 65-year-old man referred for clinical genetics evaluation for a personal and familial history of gastric cancer. He was diagnosed with stage II intestinal type gastric cancer at 45 years of age. He was treated with platinum-based adjuvant chemotherapy after surgery. He had local lymph node recurrence after two years, and subsequently underwent folinic acid, fluorouracil, and irinotecan (FOLFIRI) chemotherapy followed by radiation therapy cycles. At the time of genetic counseling, he was free of local and distant disease. He reported a sister (II:2) deceased due to diffuse gastric cancer at age 30, with an unaffected 44-year-old daughter (III:1). He also reported a 70-year-old sister (II:6) who had developed gastric diffuse cancer at age 52 and ovarian serous cancer at age 54. The parents of the proband died from chronic obstructive pulmonary disease, the father (I:1) at age 77 and the mother (I:2) at age 71. The family medical history could not be traced back further. Written informed consent was gathered for the molecular analysis of a panel of cancer susceptibility genes related to DNA damage repair and cell cycle control (*APC*, *ATM*, *BARD1*, *BRIP1*, *CDH1*, *CDK4*, *CDKN2A*, *CHEK2*, *EPCAM*, *MLH1*, *MRE11*, *MSH2*, *MSH6*, *MUTYH*, *NBN*, *PALB2*, *PMS2*, *PTEN*, *RAD50*, *RAD51C*, *RAD51D*, *RECQL1*, *SMAD4*, *STK11,* and *TP53)* on genomic DNA extracted from peripheral leukocytes from the proband. The next generation sequencing (NGS) analysis was performed on an Ion Personal Genome Machine (Ion PGM™) platform (Thermo Fisher Scientific, Carlsbad, CA, USA). The Torrent suite tools were used for analysis (Version 5.10, Thermo Fisher Scientific, Carlsbad, CA, USA). Variant classification was carried out according to the American College of Medical Genetics and Genomics (ACMG) recommendations [13]. Variant validation and segregation studies were performed by capillary electrophoresis Sanger sequencing. The identified variant was reported in accordance with the Human Genome Variation Society nomenclature guidelines (https://varnomen.hgvs.org/ last accessed on 26 June 2023). All first-degree relatives were offered genetic counseling, and written informed consent was gathered before undergoing the segregation test. The study was approved from the Ethical Committee “Comitato Etico Territoriale Lazio Area 1”, protocol 565/2023, 21 July 2023.

## 3. Results

Molecular testing identified the heterozygous c.5944C>T, p.(Gln1982*) variant in the *ATM* gene (MIM *607585; NM_000051.3; NP_000042.3). Sanger sequencing confirmed the occurrence of the variant in heterozygosity in the proband (II:3), his sister (II:6), and his niece (III:1), indirectly confirming the carrier status of the deceased individual (II:2, Figure 1). The variant is very rare, with an allele count of zero in the GnomAD population database (both v2.1.1 and v.3.1.1). It is reported in literature as “Pathogenic” and is associated with Ataxia Telangiectasia in homozygosity [14]. It can be classified as “Pathogenic” according to the American College of Medical Genetics and Genomics guidelines [13]. The pedigree of the family, along with the molecular testing results, are presented in Figure 1.

## 4. Discussion

The *ATM* gene encodes for a multifunctional serine/threonine kinase, which acts as a double-strand break DNA damage sensor [15]. It is considered as the master regulator of the DNA double strand break (DSB) response pathway, which mediates DNA damage repair, cell cycle regulation, and apoptosis [15]. The ATM protein homodimer is recruited to DSBs by the Mre11-Rad50-Nbs1 (MRN) complex and is thereby divided into two ATM monomers. These interact with DNA free ends and recruit further pathway effectors [16]. The common pathway of DNA damage sensing and cell replication control is shared with other proteins, such as BRCA1, PLAB2, BRCA2, and RAD51, whose genes are implied in cancer susceptibility conditions [15].

Biallelic pathogenic variants (in homozygosity or compound heterozygosity) in the *ATM* gene (Ataxia Telangiectasia Mutated; MIM *607585) cause Ataxia Telangiectasia (MIM #208900), an autosomal recessive condition characterized by cerebellar ataxia, immunodeficiency, skin and mucosal telangiectasias, radiosensitivity, and susceptibility to leukemia and lymphoma [17].

Heterozygous *ATM* pathogenic variants are known to result in significant breast cancer susceptibility and confer an increased risk for gastric, pancreatic, colorectal, ovarian, and prostate cancers (and susceptibility to breast cancer; MIM #114480) [17].

The study of these variants is hindered by their frequency in the general population, which ranges from 0.3% to 1% [11,18] and the reduced penetrance of the cancer susceptibility and the overall population frequency of the associated malignancies.

A possible association between germline *ATM* variants and an increased risk for gastric cancer had initially been suggested in genome-wide association studies (GWAS) [10], as well as in reports of individual families with high gastric cancer rates segregating with an *ATM* variant [19]. The frequency of deleterious *ATM* variants in patients with gastric cancer was thereby demonstrated to be significantly higher than in controls [19]. Apparently, there is no correlation between the presence of *ATM* PVs and the onset of a specific gastric cancer histotype as the association with *ATM* PVs has been described for both diffuse- and intestinal-type gastric adenocarcinoma [10]. Recently, in a cohort of 282 Chinese patients with gastric adenocarcinoma tested for germline variants in a panel of 69 cancer susceptibility genes, *ATM* PVs were the most common, with a prevalence of 1.1% in affected individuals [20]. All identified *ATM* PVs were truncating (nonsense or frameshift variants) and were associated with a lower age of onset (mean age of 49.3) compared to the age of all other patients with pathogenic variants in other genes (mean age of 58.5) or to patients with no PVs from the same cohort (mean age of 60.5) [20]. In comparison, the worldwide median age of onset for gastric cancer is approximately 70 years of age [21]. In a previous Icelandic study, the mean age of onset in the general population was 69.6 years, and the estimated effect of *ATM* variants on the age of onset was −6.1 years [10].

Currently, the penetrance for each tumor is still not fully ascertained, but odds ratios for specific malignancies in *ATM* variant carriers are emerging [11]. The NCCN guidelines suggest surveillance for breast cancer in all female patients and recommends ovarian and pancreatic screening in *ATM* PV carriers with positive family histories [12]. Screening for gastric cancer is not included in the recommendations. Other prevention options are left to the clinician’s choice. A recent large-scale study aiming to score specific risks for heterozygous *ATM* variant carriers demonstrated moderate-to-high risks for pancreatic, prostatic, gastric, and female invasive ductal breast malignancies and low-to-moderate risks for breast ductal carcinoma in situ, male breast cancer, ovarian cancer, colorectal cancer, and melanoma [11]. The authors also suggested that in limited instances, genotype–phenotype assumptions might be proposed for some mutations, tying specific risks to single variants [11]. These data, along with the mean earlier onset of gastric cancer in *ATM* carriers compared to the general population [20], support considering specific surveillance in individuals and families harboring heterozygous *ATM* PVs, at least in cases with positive family histories.

Despite the growing evidence, there are only few detailed clinical reports of families with *ATM* PVs displaying higher-than-expected gastric cancer occurrences [19]. Such reports are important as they might suggest whether individual mutations result in specific risks or suggest possible surveillance patterns. Even if genotype–phenotype assumptions are not established, the high familial occurrences of specific malignancies should always prompt the consideration of personal and familial histories when defining oncology surveillance for individuals harboring such mutations, with personalized and tailored approaches.

Familial early-onset gastric cancer with high penetrance is an unusual presentation for *ATM* variants. In the family reported on in this study, three siblings were affected by gastric cancer, of which two were early onset cases (before 45 years of age). One of the siblings also presented serous ovarian cancer. No history of AT or of other *ATM*-related malignancies was reported in their parents or other individuals. This is not uncommon in families with cancer susceptibility variants as for most forms, the penetrance for each malignancy type is far from complete, even when the risk is moderate-to-high [11]. It is to be noted that while the occurrence of gastric cancer in the three siblings can be ascribed to the *ATM* PV, the possible contribution of environmental factors or other genetic variations cannot be excluded. In particular, the possible co-occurrence of variants in other susceptibility genes that were not featured in the panel performed, such as the *CTNNA1* gene, which is associated with hereditary diffuse gastric cancer [22], is worth mentioning. The c.5944C > T, p.(Gln1982*) variant detected in this family has previously been described in a single case in homozygosity in a Saudi patient with Ataxia Telangiectasia [14]. No information was provided about family history in the original report. Usually, homozygous and compound heterozygous nonsense and frameshift deletions in *ATM* result in more severe Ataxia Telangiectasia phenotypes and account for most of the variants, while less deleterious missense variants appear to be less common among pathogenic alleles and result in milder clinical pictures [14]. Genotype–phenotype correlations for the cancer susceptibility phenotypes are more elusive. For most cancers and variants, no assumptions can be made. It appears the c.7271T>G, p.(Val2424Gly) missense variant might be associated with a higher risk for breast cancer compared to other PVs, possibly due to a dominant negative effect [11]. As in other cancer susceptibility genes, the heterozygous germline *ATM* variants identified in a cohort of patients with gastric cancer were truncating [20]. We cannot exclude that the nonsense *ATM* variant identified in this family might be correlated with a specifically higher gastric cancer risk, but further research is needed to provide an assertion. The population frequency of *ATM* PV carriers and of sporadic gastric cancer cases, the moderate and age-dependent penetrance of the cancer susceptibility phenotype for *ATM*, and the rarity of the identified variant pose significant challenges to genotype–phenotype correlation attempts. Ultimately, it cannot be stated whether the peculiar presentation of the reported family occurred by chance, was due to specific variant-related risks, or was the result of other non-investigated factors.

## 5. Conclusions

The family discussed herein shows how a defined clinical presentation in clinical cancer genetics can lead to unexpected molecular findings. These findings propose multiple significant counseling challenges. Genetic counseling should consider both the malignancy risk and the chances of Ataxia Telangiectasia in offspring given the high frequency of heterozygous carriers in the general population. Considering the conservative guidelines and the lack of data on specific malignancy risks and genotype–phenotype correlation, clinical geneticists and onco-geneticists should opt for tailored comprehensive approaches that integrate guidelines, molecular data, and individual and family histories. The high penetrance for gastric cancer in this family and the tendency towards early-onset malignancies in *ATM* carriers inferred from the literature support a possible role for gastric cancer surveillance, at least in cases with positive family histories.

## Figures and Tables

**Figure 1 biomedicines-11-02062-f001:**
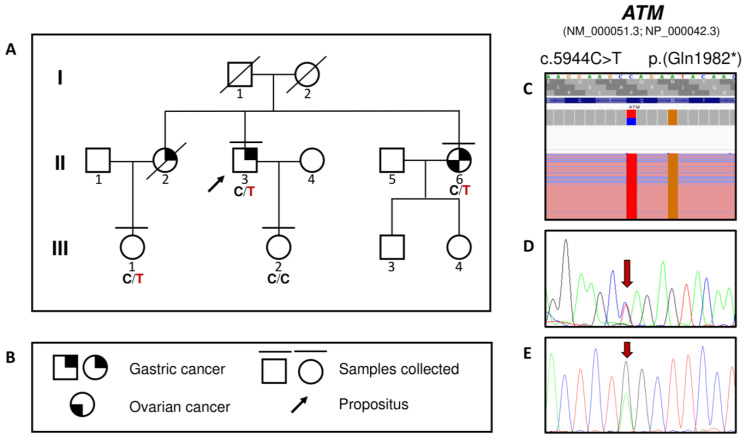
(**A**) Pedigree of the family. The proband (II:3) presented gastric cancer at age 45. His sister II:2 died at age 30 due to gastric cancer, while II:6 presented gastric cancer at age 52 and ovarian cancer at age 54. (**B**) Figure legend. (**C**) Visualization of the c.5944C > T ATM variant on an Integrative Genome Viewer (IGV). The BAM file is from the next generation sequencing experiment performed on II:3. (**D**) The Sanger validation of the variant (forward strand). (**E**) The Sanger validation of the variant (reverse strand).

## Data Availability

All data are available upon reasonable request from the corresponding author.

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
