# Peer review of "Heterozygous Pathogenic Nonsense Variant in the ATM Gene in a Family with Unusually High Gastric Cancer Susceptibility"

_biomedicines, 2023, doi:10.3390/biomedicines11072062_

Round 1

Reviewer 1 Report

The case report by Daniele Guadagnolo et al., entitled Heterozygous pathogenic nonsense variant in the ATM gene resulting in unusually high gastric cancer susceptibility, presents an important and useful study on molecular testing of the ATM gene.

The manuscript covers a family of three siblings that presents an ATM mutation considered responsible for causing gastric cancer. The manuscript is well-written and exposed and meets all the publishing criteria of a case report. Also, the authors exposed important information found in the present literature and developed a complete discussion section comprising the personal results in comparison to already published data. Also, the limitations of the actual published data were taken into consideration, underlying the relevance of exploring more the ATM mutations in order to help understanding better the cancer molecular mechanism.

I find the case report very useful for oncological research.

The manuscript is well-written and exposed and meets all the publishing criteria of a case report.

Author Response

The case report by Daniele Guadagnolo et al., entitled Heterozygous pathogenic nonsense variant in the ATM gene resulting in unusually high gastric cancer susceptibility, presents an important and useful study on molecular testing of the ATM gene.

The manuscript covers a family of three siblings that presents an ATM mutation considered responsible for causing gastric cancer. The manuscript is well-written and exposed and meets all the publishing criteria of a case report. Also, the authors exposed important information found in the present literature and developed a complete discussion section comprising the personal results in comparison to already published data. Also, the limitations of the actual published data were taken into consideration, underlying the relevance of exploring more the ATM mutations in order to help understanding better the cancer molecular mechanism.

I find the case report very useful for oncological research.

The manuscript is well-written and exposed and meets all the publishing criteria of a case report.

Dear reviewer,

to thank you for the appreciation you showed towards our work and for taking the time to thoroughly review the manuscript.

Reviewer 2 Report

This interesting manuscript describes a small family group with 3 siblings who all developed gastric cancer and who each carry a truncating ATM mutation. The manuscript itself is very well written. English language is excellent, the single image is informative, and the scientific content is of a very high standard. The authors have also summarised background literature and highlighted gaps in our knowledge as well as the difficulties associated with research and clinical management of very rare conditions such as this. I recommend this manuscript for publication.

There is an area of the subject that I feel has not been covered in the discussion and this is the concept that the unusually high penetration of gastric cancer in the 3 siblings and the young age at onset may be due to contributions of other genetic variants (in combination with the ATM mutation). Neither parent of the 3 siblings developed gastric cancer during their lifetime and yet, because all 3 siblings carried the same ATM mutation, it appears that the mutation would have been inherited from one of the parents (i.e. it is not a de novo mutation).   [Is it proven that these are the biological parents of the siblings? That may be a difficult question to answer without permission for genetic testing and archival material to test.]

It is also notable that while a number of cancer susceptibility genes were sequenced in affected family members, the screen is not comprehensive and several rare cancer susceptibility genes including a number of cancer susceptibility genes not previously associated with gastric cancers were not sequenced. This list includes a known gastric cancer susceptibility gene, CTNNA1. Mutations or variants in other genes that on their own might not be recognised as cancer susceptibility genes may also increase the penetrance of the ATM mutation and the gastric cancer susceptibility in the three siblings.

Although it is not mandatory, I would prefer to see a short paragraph added to the Discussion section of the manuscript to address these points, as I feel that it will leave scope for additional interpretations of the findings and perhaps encourage broader studies on this family and on other families with inherited cancers that are characterised by unusual presentations.

Author Response

This interesting manuscript describes a small family group with 3 siblings who all developed gastric cancer and who each carry a truncating ATM mutation. The manuscript itself is very well written. English language is excellent, the single image is informative, and the scientific content is of a very high standard. The authors have also summarised background literature and highlighted gaps in our knowledge as well as the difficulties associated with research and clinical management of very rare conditions such as this. I recommend this manuscript for publication.

Thank your for your appreciation.

There is an area of the subject that I feel has not been covered in the discussion and this is the concept that the unusually high penetration of gastric cancer in the 3 siblings and the young age at onset may be due to contributions of other genetic variants (in combination with the ATM mutation). Neither parent of the 3 siblings developed gastric cancer during their lifetime and yet, because all 3 siblings carried the same ATM mutation, it appears that the mutation would have been inherited from one of the parents (i.e. it is not a de novo mutation).   [Is it proven that these are the biological parents of the siblings? That may be a difficult question to answer without permission for genetic testing and archival material to test.

It is also notable that while a number of cancer susceptibility genes were sequenced in affected family members, the screen is not comprehensive and several rare cancer susceptibility genes including a number of cancer susceptibility genes not previously associated with gastric cancers were not sequenced. This list includes a known gastric cancer susceptibility gene, CTNNA1. Mutations or variants in other genes that on their own might not be recognised as cancer susceptibility genes may also increase the penetrance of the ATM mutation and the gastric cancer susceptibility in the three siblings.

Although it is not mandatory, I would prefer to see a short paragraph added to the Discussion section of the manuscript to address these points, as I feel that it will leave scope for additional interpretations of the findings and perhaps encourage broader studies on this family and on other families with inherited cancers that are characterised by unusual presentations.

Thank you for your observations.

As you assume, parental analysis could not be performed.

We agree that a more comprehensive presentation of the limitations you disclosed would be beneficial for the article.

The mentioned limitations are now detailed in the first sentences of the last paragraph of the discussion. the section now reads:

“Familial early-onset gastric cancer with high penetrance is an unusual presentation for ATM variants. In the family reported in this study, three siblings were affected by gas-tric cancer, of which two were early onset cases before 45 years of age. One of the sibling also presented serous ovarian cancer. No history of AT or of other ATM-related malignan-cies was reported in their parents or other individuals. This is not uncommon in families with cancer susceptibility variants, as for most forms the penetrance for each malignancy type is far from complete, even when the risk is moderate-to-high [11]. It is to be noted that, while the occurrence of gastric cancer in three siblings can be ascribed to the ATM PV, the possible contribution of environmental factors or other genetic variations cannot be excluded. In particular, the possible co-occurrence of variants in other susceptibility genes that are not featured in the panel performed, such as the CTNNA1 gene associated with hereditary diffuse gastric cancer [22], is worth mentioning. “

Reviewer 3 Report

The authors report that a heterozygous pathogenic nonsense variant in the ATM gene may result in especially high gastric cancer susceptibility. The inherent question concerning such reports is whether the described observations occurred by chance. There must be a high number of families heterozygous for ATM variants worldwide and the question is if the specific reported variant results in higher risk than other variants. This should be mentioned clearer in a “limitations” paragraph before the Discussion.

However, the Discussion in general is well written and answers many questions, including the lack of association between ATM variants and specific gastric cancer variants which should be expected given its role in DNA damage repair. It has previously been reported that the average age of onset in ATM mutation carriers was 49 years (vs 58), referring to such numbers is important. Please refer to the age of onset in carriers and non-carriers in European populations if available.

Minor comments:

Line 64: Reference 12 does not seem to match the statement in the text, please check.

In line 163 it is mentioned that three siblings had early onset gastric cancer. However, the common definition of early onset is 45 years or younger. Please rephrase.

Author Response

The authors report that a heterozygous pathogenic nonsense variant in the ATM gene may result in especially high gastric cancer susceptibility. The inherent question concerning such reports is whether the described observations occurred by chance. There must be a high number of families heterozygous for ATM variants worldwide and the question is if the specific reported variant results in higher risk than other variants. This should be mentioned clearer in a “limitations” paragraph before the Discussion.

Thank you for your observation. We agree that a more thorough presentation of the limitations you mentioned was needed. However, due to the structure of the article and to the integrations suggested by other reviewers, we preferred to present these considerations in the final paragraph of the discussion.

We added the following sentences at the end of the discussion:

“The population frequency of ATM PV carriers, and of sporadic gastric cancer cases, the moderate and age-dependent penetrance of the cancer susceptibility phenotype for ATM, and the rarity of the identified variant pose significant challenges to genotype-phenotype correlation attempts. Ultimately, it cannot be stated whether the peculiar presentation of the reported family occurred by chance, due to specific variant-related risks, or as a result of other non-investigated factors. “

However, the Discussion in general is well written and answers many questions, including the lack of association between ATM variants and specific gastric cancer variants which should be expected given its role in DNA damage repair. It has previously been reported that the average age of onset in ATM mutation carriers was 49 years (vs 58), referring to such numbers is important. Please refer to the age of onset in carriers and non-carriers in European populations if available.

Thank you for your appreciation and for your comments. Data on the median worldwide onset of gastric cancer, along with data on the earlier onset in carriers of ATM variants in an European population are now provided (with references).

We added:

“By comparison, the worldwide median age of onset for gastric cancer is about 70 years of age [21]. In a previous Icelandic study, the mean age of onset in the general population was 69.6 years, and the estimated effect of ATM variants on the age of onset was −6.1 years [10].”

Minor comments:

Line 64: Reference 12 does not seem to match the statement in the text, please check.

The reference has been replaced with the most recent online version of the guidelines.

In line 163 it is mentioned that three siblings had early onset gastric cancer. However, the common definition of early onset is 45 years or younger. Please rephrase.

The sentence has been reworded as:

“In the family reported in this study, three siblings were affected by gastric cancer, of which two were early onset cases (before 45 years of age)”

Reviewer 4 Report

The data presented are of interest to oncologists. Figures are appropriate. 

The relationship between the PV in the ATM gene and gastric cancer susceptibility is described as “resulting in” (line 2), “due to” (line 20),  “implied in”( line 37), “the role of” (line 132), “correlation with or to” (line 136, line 178), “associated with” (line 174). A better explanation and more uniform terminology is warranted.

Author Response

The data presented are of interest to oncologists. Figures are appropriate. 

The relationship between the PV in the ATM gene and gastric cancer susceptibility is described as “resulting in” (line 2), “due to” (line 20),  “implied in”( line 37), “the role of” (line 132), “correlation with or to” (line 136, line 178), “associated with” (line 174). A better explanation and more uniform terminology is warranted.

Thank you for your  appreciation and for your observations. We edited the sentences with the mentioned expression to better clarify the type of association we are referring to in each phrase

Line 2:  the title now reads “Heterozygous pathogenic nonsense variant in the ATM gene in a family with unusually high gastric cancer susceptibility.”

Line 20: the statement is now less assertive and reads “We describe a family presenting early-onset gastric cancer, harboring an heterozygous Pathogenic ATM variant. “

Line 37: the sentence has been reworded as “The genetic landscape of monogenic cancer susceptibility is varied and complex, with many variants in several genes possibly involved in determining the susceptibility to different neoplasms”

Line 132: the sentence now reads “A possible association between germline ATM variants and an increased risk for gastric cancer has been initially suggested by Genome-Wide Association Studies”

Line 136 (see below for 178) “correlation with”: the sentence has been edited to better explain the information retrieved from the cited source

Line 174 “associated with” is the  appropriate term in this case.

Line 178-179 “With” is the correct preposition in this case. The text has been edited.